# Selected Plant-Derived Polyphenols as Potential Therapeutic Agents for Peripheral Artery Disease: Molecular Mechanisms, Efficacy and Safety

**DOI:** 10.3390/molecules27207110

**Published:** 2022-10-21

**Authors:** Guglielmina Froldi, Eugenio Ragazzi

**Affiliations:** Department of Pharmaceutical and Pharmacological Sciences, University of Padova, 35131 Padova, Italy

**Keywords:** baicalein, curcumin, mangiferin, quercetin, resveratrol, diabetes mellitus, endothelial dysfunction, peripheral artery disease, polyphenols, cardiovascular diseases

## Abstract

Vascular diseases, such as peripheral artery disease (PAD), are associated with diabetes mellitus and a higher risk of cardiovascular disease and even death. Surgical revascularization and pharmacological treatments (mainly antiplatelet, lipid-lowering drugs, and antidiabetic agents) have some effectiveness, but the response and efficacy of therapy are overly dependent on the patient’s conditions. Thus, the demand for new cures exists. In this regard, new studies on natural polyphenols that act on key points involved in the pathogenesis of vascular diseases and, thus, on PAD are of great urgency. The purpose of this review is to take into account the mechanisms that lead to endothelium dysfunction, such as the glycoxidation process and the production of advanced glycation end-products (AGEs) that result in protein misfolding, and to suggest plant-derived polyphenols that could be useful in PAD. Thus, five polyphenols are considered, baicalein, curcumin, mangiferin, quercetin and resveratrol, reviewing the literature in PubMed. The key molecular mechanisms and preclinical and clinical studies of each selected compound are examined. Furthermore, the safety profiles of the polyphenols are outlined, together with the unwanted effects reported in humans, also by searching the WHO database (VigiBase).

## 1. Introduction

Non-communicable diseases, such as cardiovascular diseases, are continuously monitored and investigated for prevalence, pathophysiology, outcome and prevention [1]; however, the impact of peripheral artery disease (PAD) is not adequately considered. Atherosclerotic occlusive disease of the lower extremities affects approximately 10% of people in their 60s and 70s, and approximately 20% of individuals over the age of 80 years [2,3]. Patients with diabetes mellitus (DM), those who smoke, the elderly and those affected by cardiovascular diseases are mainly at risk of encountering PAD [2,4]. A calculator to predict the lifetime risk of PAD is available online [5].

There is convincing evidence to support the fact that the primary pathophysiology of PAD is triggered by the obstruction of atherosclerotic plaques of the lower extremities causing a consequent decrease in blood flow (ankle brachial index, ABI ≤ 0.9) [6]. Leg pain during exercise due to poor blood flow (intermittent claudication) is a major symptom of PAD; in fact, increasing blood supply to the compromised limb after surgical revascularization can improve symptoms and hemodynamic parameters in the patients [7]. Primarily, antiplatelet therapy is recommended to reduce cardiovascular complications in PAD, whereas β-adrenergic inhibitors should be used with caution, if clinically indicated [8]. Other agents used in patients with PAD are standard lipid-lowering therapies, mainly the administration of statins [8]. In general, several therapeutic agents can improve endothelial vasodilator function and insulin resistance: pioglitazone, metformin, angiotensin-converting enzyme (ACE) inhibitors and angiotensin II-receptor antagonists (AT1-receptor blockers) [8]. Additionally, various vasoactive drugs could be used to treat symptoms, such as improving walking distances [2]. Furthermore, propionyl-L-carnitine has been suggested to alleviate PAD symptoms through a metabolic pathway, thus improving exercise performance [9,10,11]. In addition, gene therapy using encoding genes of various types of growth factors is at the beginning stage [12].

Although these approaches have some success, the demand for other types of remedies remains unmet, and, for this reason, a large quantity of food supplements are commercially proposed, often without scientific evidence, to decrease PAD symptoms.

### 1.1. Mechanisms of PAD

Diabetes mellitus, especially if not properly treated, can cause damage to different organs and tissues of the human body. Microvascular complications include retinopathy, neuropathy and nephropathy, while macrovascular complications include cardiovascular diseases, stroke and peripheral vascular disease that can lead to bruising or lesions that do not heal, gangrene and, in severe cases, limb amputation. Peripheral artery disease (PAD), also called peripheral vascular disease, is caused by the narrowing of the lumen of blood vessels that carry blood to the arms, legs, stomach and kidneys. In people affected by DM, the risk of PAD increases with age, with the duration of diabetes and in the presence of neuropathy. Other factors associated with cardiovascular diseases, such as high C-reactive protein and homocysteine levels, are also associated with an increased risk of PAD [13,14]. Peripheral arteriopathy is characterized by two types of symptoms: intermittent claudication (or intermittent pain, which may occur during exercise or walking, but resolves with rest) and pain at rest (which is caused by chronic hypoxia/ischemia in the limb, with inadequate blood flow to the affected limb). This pathology is an important risk factor for lower-extremity amputation. Unfortunately, its diagnosis is often made too late and occurs when symptoms are marked, delaying a preventive treatment [15]. Insulin resistance, commonly manifested in type 2 diabetes mellitus (T2DM), is a consequence of several risk factors represented by obesity, sedentary behavior and aging; therefore, hyperglycemia, diabetes, hypertension and dyslipidemia are concomitant diseases often observed in patients affected by PAD [10]. In patients with DM, the diagnosis of symptomatic PAD is approximately twice as high as in patients not affected by diabetes [16]. Recently, Lilja et al. showed that men with DM developed symptomatic PAD more frequently compared to women (15.5% vs. 8.9%) [17], even if not all studies have observed significant differences in the prevalence of PAD between sexes.

Several molecular pathways are involved in the pathogenesis of arterial insufficiency, such as (i) endothelial dysfunction, e.g., the presence of increased plasma levels of asymmetric dimethylarginine (ADMA) is associated with endothelial vasodilator dysfunction, causing arterial stiffness and a reduction in NO^•^ production [18,19]; (ii) endoplasmic reticulum (ER) dysfunction, e.g., vascular endothelial cells of hyperglycemic subjects are characterized by an altered, rough endoplasmic reticulum and protein folding [20,21]; (iii) promotion of inflammation by secretion of cytokines, such as TNF–α, IL–1, IL–6 and IL–8, and chemotactic stimulus for monocytes and macrophages [22]; (iv) mitochondrial dysfunction and increased oxidative stress-induced damage, also linked to the activation of the transcription nuclear factor–κB (NF–κB) [23,24]; and (v) interaction between advanced glycation end-products (AGEs) and their receptors, causing inflammation and endothelial dysfunction [25,26].

The differences in endothelium function between women and men have been reported in various studies. The beneficial vascular effects of estrogens are related to the modification of the functional state of the endothelium [27]. The impaired endothelium-dependent vasodilation of the coronary or peripheral vasculature is positively correlated with an increased risk of cardiovascular events and is an independent predictor of vascular morbidity and mortality.

### 1.2. Role of AGEs

In conditions of persistent hyperglycemia, molecular rearrangements occur between tissue proteins and glucose or other reducing carbohydrates, leading to the irreversible formation of AGEs, which significantly contribute to the development of complications associated with DM, including PAD [28]. It has also been observed that the accumulation of AGEs worsens endothelial function [29,30].

The formation of AGEs begins with the non-enzymatic glycation of free amino groups by sugars and aldehydes, which leads to a succession of rearrangements of intermediate compounds and, finally, to irreversibly bound products known as AGEs [24]. Glycation and oxidative stress are intricately linked, and both phenomena are referred to as glycoxidation [28]. Persistent hyperglycemia and oxidative stress accelerate the formation of AGEs [31]. These have mainly been detected in long-lived proteins, with post-translational AGE modifications mainly occurring at the side-chain amino groups of the lysine and arginine residues. Glycation causes the irreversible modification of the protein structure and consequent loss of functionality, leading to detrimental effects in tissues, i.e., vasculature. Glycated proteins become less prone to proteolysis as a consequence, e.g., the accumulation of glycated collagen causes the thickening of blood vessels. Recently, authors showed that AGEs can cause the apoptosis of endothelial progenitor cells via nicotinamide adenine dinucleotide phosphate (NADPH) oxidase and promote atherosclerosis [32]. Nucleotides and lipids are also possible targets of glycation, causing the DNA mutation and modification of the integrity of the cell membrane, triggering cell death. Thus, it can be believed that compounds that counteract the effect of macromolecule damage (glycation-protective effects) can slow down the progression of vascular damage in the legs and, furthermore, also in other districts, such as the brain, protecting the health of humans [33]. Recently, several natural compounds, i.e., flavonoids and more generally polyphenols, have received attention for their endothelial protective effects [26,34].

### 1.3. Protective Activity of Polyphenols

Natural polyphenols are a heterogeneous group of secondary metabolites produced in plants that have several functions in the plant kingdom, such as the protection against pathogens, including microorganisms, insects and fungi, and protective activity against ultraviolet radiation and harmful environmental factors; moreover, they can also act as attractants of beneficial organisms [35,36]. These secondary compounds are biosynthesized through the shikimic acid and phenylpropanoid pathways and are believed to be participative in adapting plants in a stressed situation due to environmental changes [37]. These compounds, according to their chemical structures, are divided into various subclasses, such as phenolic acids, flavonoids, tannins, coumarins, lignans, quinones, stilbenes and curcuminoids [38].

New research on plant-derived polyphenols as a guide for exploring innovative synthetic compounds can provide new knowledge to develop new drugs in PAD treatment. In terms of this point, plant-derived natural products approved for therapeutic use in the last 30 years are used for the treatment of various diseases and modulate a wide range of molecular targets. Recent products approved as drugs by the FDA are Veregen^®^ (sinecatechins; green tea (*Camellia sinensis* L.) leaf extract), Fulyzaq^®^ (crofelemer; extract from the red latex of *Croton lechleri* Müll. Arg.) and Grastek^®^ (Timothy grass (*Phleum pratense* L.) pollen allergen extract). Therefore, natural products still represent an important pool for the identification of new pharmacological tools at present.

The purpose of this review is to summarize existing evidence from preclinical and clinical studies on the contribution of natural polyphenols as vasodilators and, in general, as favorable agents for blood circulation, suggesting their potential use in PAD and against an associated causative disease: diabetes. Obviously, vasodilators can be useful not only in PAD, but also in several other cardiovascular diseases, such as hypertension, angina pectoris, heart failure and others. When searching PubMed for “natural polyphenols and vasodilators” or “natural polyphenols and vasodilation” using the criteria “English language” and “full texts”, a total of 273 entries were found (23 September 2022), suggesting a wide research interest in the topic (Appendix A). However, when the including criterion “clinical trial” was applied, only six articles remained, of which five referred to the administration in healthy subjects of plant extracts, i.e., grape seed extract [39], apple extract [40], cocoa powder [41], a mixture of three plants (*Kaempferia parviflora*, *Punica granatum* and *Moringa oleifera*) [42] and a beverage containing numerous fruits and vegetables [43]. Only one trial focused on the administration of a single compound, resveratrol, administered in patients after myocardial infarction [44]. Furthermore, another search for “peripheral artery disease and natural compounds” produced only 35 items in the time interval from 1962 up to July 2022 (Appendix A). From these findings, several polyphenolic compounds have been selected and studied in detail in this review. We also wanted to highlight some observations on the importance of physical exercise, as a non-pharmacological approach, in preventing and slowing the onset of circulatory problems, partly derived from a sedentary lifestyle.

## 2. Prevention and Protective Factors: The Risk-Reduction Approach

### 2.1. Moderate Physical Exercise

Cross-sectional and longitudinal trials indicate that aerobic exercise can prevent and reverse age-associated arterial stiffness [45]. Moderate exercise tends to be a stimulator of NO^•^ release and favors endothelial function, which in turn reduces the cardiovascular risk profile in patients affected by DM [46]. Recently, a clinical trial was conducted on 33 healthy middle-aged and older subjects (67 ± 1 years), randomly divided into two groups, of which one had aerobic exercise training (AT), while the second was a sedentary control [19]. Circulating apelin and adropin levels gradually increased during the AT intervention and significantly increased from baseline at weeks 4, 6 and 8. Furthermore, plasma ADMA levels significantly decreased during the 8-week AT intervention. Therefore, this study suggested that exercise training induces favorable changes in the time course of NO^•^ production, participating in AT-induced improvements of central arterial stiffening with advancing age [19]. The data indicate that exercise training can increase endothelium-derived nitric oxide activity in patients with an impaired endothelial function [45], and positively modulates inflammation and the atherosclerotic process; consequently, these facts can attenuate the progression of lower-limb myopathy, with subsequent improvements in the patients’ functional capacity and health-related quality of life [47,48].

A multicenter observational prospective cohort study on 500 PAD patients (53% of them also affected by DM) who underwent endovascular treatment showed that the implementation of home-based exercise, monitored by pedometers, significantly decreased the risk of major adverse events, including death and the amputation of the target limb [49]. Supervised exercise therapy (SET), implemented several times a week and gradually, is, to date, an approach that should be offered to patients affected by symptomatic PAD [50], and is considered a class IA (highest level) recommendation, according to the 2016 AHA/ACC Guideline on the Management of Patients with PAD [51]. Unfortunately, the long-term participation in these programs is low and, therefore, this approach is underused.

A recent study conducted on PAD patients showed that exercise induces an exaggerated increase in arterial mean pressure compared to controls, but popliteal artery blood flow remains impaired, and concomitantly inflammatory and oxidative stress markers increase [52]. These observations, although derived from a small group of subjects (8 PAD patients), suggest that the improvement of exercise tolerance should be considered as a therapeutic target for people affected by PAD and, furthermore, suggest that the use of anti-inflammatory and antioxidant agents may be a promising approach.

### 2.2. Plant-Derived Compounds

Among natural polyphenols, flavonoids are present in most plants, having antioxidant activity. Phenols and flavonoids can directly scavenge reactive oxygen species (ROS) as a result of their ability to donate hydrogen atoms or electrons. Every plant contains a unique combination of phytochemicals, which can cause different beneficial effects in individuals. Green tea, grapes, apples, ginkgo biloba, soybean, turmeric, berries and onions are rich in flavonoids and are used as natural food supplements. Many studies also confirmed the protective effects of polyphenol antioxidants against diabetic vascular complications [53,54,55,56], such as the effects of green tea and cocoa polyphenols on endothelial dysfunction in patients with DM, red-wine polyphenols on microvascular dysfunction and citrus-fruit consumption on vascular protection [57,58]. Virgin olive oil rich in polyphenols has a cardioprotective effect suggested with a consumption between 20 and 30 g/day [59]. The ingestion of 40 g of dark chocolate in 22 heart transplant subjects improved coronary circulation and decreased platelet adhesion 2 h after supplementation [60,61]. However, the heterogeneity of the results does not allow one to make a definitive evaluation on chocolate’s usefulness in DM [62]. A randomized, double-blind, placebo-controlled cross-over trial is in development to investigate whether cocoa flavanols improve blood pressure and vascular reactivity in patients with T2DM [63]. Overall, the European Food Safety Authority (EFSA) stated that 200 mg daily of cocoa flavanols (10 g of high-flavanol dark chocolate) could be beneficial for endothelium-dependent vasodilation in healthy populations [64]. Therefore, there are several epidemiological studies and various prospective clinical trials suggesting that a diet rich in polyphenols reduces the risk of incurring in cardiovascular diseases, but a clinical evaluation of specific phenols is lacking.

The present proposal focused on the study of the activity of five selected plant-derived polyphenolic compounds in the treatment of vascular damage, especially linked to hyperglycemia. With this aim in mind, the selected polyphenols investigated were baicalein, curcumin, mangiferin, quercetin and resveratrol (Figure 1). Of course, to plan a factual intervention against vascular damage, particularly with respect to PAD, known risk factors, including smoking, hyperglycemia, hypertension, dyslipidemia, chronic kidney disease and depression, should be identified and limited as much as possible.

## 3. Specific Polyphenolic Compounds of Interest

The existing knowledge from preclinical and clinical studies, as well as the recognized side effects, on the selected plant-derived polyphenols considered in this review are summarized in Table 1 and Table 2.

### 3.1. Baicalein

#### 3.1.1. Chemistry and Sources

Baicalein (5,6,7–trihydroxyflavone) is a naturally occurring polyphenol found in several medicinal plants, in particular in the roots of *Scutellaria baicalensis* G. and *Scutellaria lateriflora* L., and in the seeds, leaves, fruits and bark of *Oroxylum indicum* (L.) Kurz [65,66], all species that grow in several Asian countries. Baicalein is the aglycone of baicalin (baicalein 7-O-glucuronide), which is also found as such in plants and, moreover, is formed after the biotransformation of baicalein in vivo [65,67].

#### 3.1.2. Activity: Preclinical Studies

Baicalein and baicalin, and extracts containing them, are proposed for various pharmacological effects, such as anti-inflammatory, antiviral, antibacterial, anticancer, antineurodegenerative and protective, against cardiovascular diseases [65,68,69]. The molecular mechanisms of action of baicalein, as well as those of other flavonoids, are mainly linked to the antioxidant activity, which occurs in different steps of the oxidative process; for example, it has demonstrated a role as scavengers of free radicals already formed in the medium (including lipid peroxyl radicals), as chelators of metal ions and by removing oxidatively altered biomolecules [70,71]. The role of the OH groups in polyphenols is dependent on the global geometry of the molecule and interactions among neighbor groups; a detailed conformational analysis and linked antioxidant mechanisms have been recognized for baicalein [70]. Recent studies on the structure–activity relationship suggest that the baicalein moiety is relevant for bioactivities [72]; however, the molecular mechanisms of baicalein are multiple and follow many molecular pathways that require further evaluation. The data confirm the beneficial role of *Scutellaria baicalensis,* which is widely used in traditional Chinese medicine to treat hypertension, respiratory infections, inflammation and diarrhea [73,74]. A study using three structurally related polyphenols, such as baicalin, baicalein and wogonin, showed the inhibition of endothelial cell barrier disruption, suggesting their protective activity against vascular inflammatory diseases [75]. Recently, studies have suggested that baicalein exhibits potential antidiabetic activities in metabolic syndrome [76]; the effect is related to the inhibition of α–glucosidase activity [77,78].

The in vitro and in vivo data suggest that baicalein is able to reduce vascular inflammation induced by high glucose levels. In human umbilical vein endothelial cells (HUVECs), baicalein (5–10 µM) was able to protect the cells from membrane disruption caused by a 25 mM glucose concentration; the polyphenol (10 µM) reduced the expression of the chemokines MPC–1 and IL–8, as well as ROS formation [75]. Furthermore, the alteration of vascular permeability induced by high glucose administration in mice was counteracted by baicalein (4.5–8.9 µg/mouse, i.v.) [75]. The vascular protective effect of baicalein was also demonstrated in vitro and in vivo as mediated by the inhibition of the high-mobility group box 1 (HMGB1) signaling pathway [79]. The role of baicalein on vascular function was assessed further by an in vitro study on the hybridoma endothelial cell line EA.hy926 [80]. The results indicate that baicalein is able to exploit protective effects against the oxidative stress of the endothelium, which is linked to the risk of diabetic angiopathy.

Baicalein has hypotensive effects on hypertensive rats, improving blood pressure and endothelium function [81]. Zhang et al., using streptozotocin and a high-fat-diet-induced diabetic rats, demonstrated that a 4–week treatment with baicalein (150 mg/kg/day) can reduce the level of blood glucose and improve insulin resistance, dyslipidemia and inflammation [82]. The effects have been attributed to the modulation of the gut microbiota, leading to increased levels of short-chain fatty acids (including acetate, propionate and butyrate), which are capable of improving gut barrier activity by stimulating epithelial growth and innate reactivity to invading bacteria [82].

A computational study revealed that baicalein is among the most promising candidates for the development of useful flavonoid derivatives in the treatment of DM [83]. Bioactivity is believed to derive from the structure that allows free-radical scavenging properties, reduces oxidized compounds, chelates metals and inhibits enzymes [83]. The experimental data suggest that baicalein can reduce oxidative stress, the expressions of iNOS and TGF–β1, as well as counteracting NF–κB activation [84]. Through a molecular modeling approach, together with microscopic and spectroscopic analyses, baicalein has also been proven to contrast the formation of AGEs and amyloid fibrils, phenomena that contribute to the loss of protein function and are connected to tissue damage [24].

In LPS-stimulated HUVECs, baicalein inhibits the expression of inflammatory cytokines IL–lβ, IL–6 and TNF–α, as well as monocyte chemoattractant protein 1 (MCP–1) [85]. The authors also demonstrated that the inhibitory activity of baicalein occurs through the TLR4/NF–κB signaling pathway. The toll-like receptor 4 (TLR4)/NF–κB cascade pathway has been considered as a pivotal mechanism that leads to endothelial inflammation, activated by inflammatory signals, such as bacterial toxins; the fact that this pathway was modulated by baicalein provides an explanation for its potential as an anti-inflammatory agent.

Huan et al. indicated that baicalein exerts an anti-angiogenic effect on the inflammation microenvironment by inhibiting the transcriptional activity of activator protein-1 (AP–1) [86]. Tsai et al. observed that baicalein inhibits the expression of the lectin-like oxidized-LDL receptor 1 (LOX–1) protein in HUVECs, therefore protecting against oxidized-LDL atherogenic effects [87]. In this experimental model, baicalein also reduced ROS formation determined by the exposure to oxidized-LDL, as well as the consequent inflammation, by modulating AMPK/PKC/NADPH oxidase/NF–κB signaling [87].

However, Machha et al. demonstrated that acute exposure to baicalein alters the vascular tone in isolated rat aorta, due to the inhibition of endothelium-derived nitric oxide [88]. A dual effect of baicalein has been suggested, evaluating the endothelium-dependent contraction versus a direct relaxation produced by the substance in a rat mesenteric artery [89]. Conversely, the chronic oral administration of baicalein improves endothelium-dependent relaxation in spontaneously hypertensive rat aorta [90]. Considering vascular protection, baicalein (10 mg/kg/day for two weeks, orally) was shown to attenuate intimal hyperplasia in an in vivo model of vascular injury induced in the carotid artery of a rat [91]. As demonstrated in rat vascular smooth-muscle cells, the effect was due to the inhibition of proliferation via the MAPK, NF–κB, PI 3–kinase pathways, and the interaction with cell cycle machinery [91]. Using RAW264.7, HUVEC and MOVAS cells, Zhang et al. observed the anti-inflammatory activity of baicalein, attributed to the activation of the AMPK/Mfn–2 axis, together with the inhibition of downstream MAPKs/NF–κB signaling transduction [84].

Vascular calcification, often observed in patients with hypertension, atherosclerosis and DM, was also evaluated as a target of the action of baicalein [92]. Experiments on primary rat vascular smooth-muscle cells (VSMCs) indicated that baicalein decreased the mineralization rate, as well as calcium deposition and alkaline phosphatase activity [92]. Furthermore, Runt-related transcription factor 2 (Runx2, a transcription factor associated with osteoblast differentiation, also a regulator of the calcification of vascular smooth-muscle cells) and bone morphogenetic protein 2 (BMP–2, an osteogenic protein implicated in vascular calcification) expressions were negatively regulated in calcified VSMCs treated with baicalein. In vivo experiments, performed on a rat model of vascular calcification, demonstrated that baicalein was capable of inhibiting vascular calcification through multiple mechanisms, including the prevention of apoptosis, suppression of Runx2–BMP–2 signaling pathways and the preservation of the vascular contractility phenotype through the increased production of α–SM22 and α–SMA (vascular smooth-muscle markers) [92].

#### 3.1.3. Activity: Clinical Studies

The pharmacokinetic characteristics of baicalein administered orally were evaluated in a cohort of 36 healthy Chinese subjects [93]. The study was a Phase-I single-center, randomized, double-blind, placebo-controlled, multiple ascending dose trial in which baicalein was administered at doses of 200, 400 and 600 mg once daily on days 1 and 10, 3 times daily on days 4 to 9. The absorption of the drug was rapid, with peak plasma levels evident within 2 h after administration. Treatment was found to be safe and well-tolerated; adverse events were mild and spontaneously resolved. Another study, consisting of multiple phases, considered a multiple design involving a total of 110 subjects, including a randomized, double-blind, placebo-controlled, single ascending dose study with doses of baicalein ranging from 100 to 800 mg [94]. In addition to the pharmacokinetic evaluation, the effect of food on the disposition of the drug was considered. The published clinical studies, although limited in sample dimension, indicate the favorable bioavailability and a good safety profile of baicalein, when used at suggested doses. However, to date, no studies on clinical effectiveness in the cardiovascular field are available.

#### 3.1.4. Safety Profile

Baicalein is considered safe and has a good tolerability profile in humans, as per se it did not have adverse reports in VigiBase [95], the WHO global database of potential side effects of medicinal products.

Mild side effects were reported in a placebo-controlled study in which 68 healthy subjects were treated with a single dose of baicalein orally, 100 to 800 mg [94]. The most common side effects were increased levels of the C-reactive protein and triglycerides, and proteinuria; these undesirable effects did not depend on the dose of baicalein administered [94]. In a clinical trial, multiple doses of baicalein (200, 400 or 800 mg twice daily) in healthy volunteers produced various adverse effects, such as abdominal pain, constipation, and increased alanine transaminase and aspartate aminotransferase levels [96].

In fact, some concerns arose from reports on the possible liver toxicity of the use of plant extracts rich in flavonoids. Recently, it was shown in vitro that baicalein can inhibit human UDP-glucuronosyltransferases1A1, the enzyme that is primarily responsible for glucuronidation and the elimination of bilirubin [97], which can cause jaundice and severe liver disease. In general, further clinical trials involving a higher number of enrolled subjects are needed to produce a clearer picture of the safety profile of baicalein in humans.

### 3.2. Curcumin

#### 3.2.1. Chemistry and Sources

Curcumin, also known as diferuloylmethane [(1*E*,6*E*)-1,7-bis(4-hydroxy-3-methoxyphenyl)hepta-1,6-diene-3,5-dione], is a diarylheptanoid that represents the main component of the turmeric rhizome (*Curcuma longa* L. and other *Curcuma* spp.), and has caused extensive investigations from biological and chemical points of view [98]. Turmeric is a widely used medicinal plant in Asian countries, appreciated for its antioxidant and anti-inflammatory activities, as well as for other multiple properties [99].

#### 3.2.2. Activity: Preclinical Studies

Curcumin inhibits proinflammatory cytokines, TNF–α, ICAM–1, NOX2 and cyclo–oxygenase–2 expressions, and reduces the leucocyte-endothelium interaction in diabetes-induced rat vascular inflammatory models [100,101]. Curcumin has been widely evaluated with respect to the modulation of endothelial function [102]. Curcumin also presents a neuroprotective potential in the pathogenesis of Alzheimer’s disease [103,104,105]. The many described biological activities of curcumin have been linked to its chemical instability that leads to the rapid autoxidation responsible for its antioxidant behavior. The oxidation of curcumin produces intermediate metabolites that conduct to a final bicyclopentadione [106,107]. Moreover, it has been demonstrated that electrophilic metabolites can covalently adduct to cellular protein targets, in particular those of the NF–κB pathway, which explains the anti-inflammatory activity of the natural compound. The methoxy groups of the molecule have also been shown to possess a role in its anti-inflammatory biological activity [108].

Ischemia and reperfusion injury in skeletal muscles, obtained by clamping the femoral artery and vein, is an experimental model that can provide information on the damage occurring in ischemic events of the limbs. Rats subjected to 4 h femoral occlusion followed by 2 h reperfusion, which received curcumin (100 mg/kg i.p., 1 h before reperfusion), presented a preservation of superoxide dismutase and catalase activities, as well as of the glutathione content in muscle tissues, compared to injured but not treated animals [109]. In the same context, the muscle and plasma levels of malondialdehyde and protein oxidation were reduced, as well as the plasma levels of inflammatory cytokines (TNF–α and IL–1β). The data suggest that curcumin has protective activity against the ischemic damage of skeletal muscles [109].

Using a mouse model of hindlimb ischemia obtained by the ligation of the left femoral artery, Liu et al. observed that curcumin (100 mg/kg, i.p. administration 1 h before the ligation surgery) is capable of improving the running capacity on a treadmill of injured mice compared to the controls, evaluated 1 and 2 weeks following injury [110]. A histological examination of skeletal-muscle sections from the curcumin-treated group showed less muscle degeneration and fibrosis than in the control mice; also, curcumin treatment was able to significantly reduce macrophage infiltration and ischemia- induced inflammation. The regulation of the tissue inflammatory response has been associated with a decrease in the proinflammatory cytokines TNF–α, IL–1β and IL–6 as a result of the diminished expression of NF–κB, namely, of the subunit p65 [110]. These data suggest that curcumin could be a promising agent used against diseases of the ischemic limbs. Another in vivo experiment conducted on streptozotocin-induced diabetic mouse that received hindlimb ischemic surgery demonstrated that curcumin (1000 mg/kg in olive oil, applied by gavage for 14 days) can promote the significant recovery of blood flow detected by laser Doppler imaging, 7 and 14 days after ischemic surgery; histology evidenced neogenic capillary formation, as a consequence of improved angiogenesis and the proliferation of endothelial progenitor cells [111]. Zhang et al. studied the role of curcumin in an experimental PAD model of murine hindlimb ischemia obtained with surgical ligation and the excision of the femoral artery. Mice received 1000 mg/kg of curcumin (in 300 µL of olive oil, gavage) once a day for two weeks [112]. In animals treated with curcumin, the laser Doppler perfusion imaging system showed perfusion recovery 14, 21 and 28 days after ischemia induction, significantly better than in the control. Histology of the ischemic gastrocnemius muscle after 28 days revealed a significant increase in capillary density. Furthermore, 7 days after surgery, a 5-fold increase in the expression of microRNA (miR)–93 (a mediator of neovascularization in PAD) was detected in the muscle tissue, suggesting the role of curcumin in angiogenesis [112].

Another possible favorable mechanism of action of curcumin in the damage of large arteries could be the inhibition of hypoxia-inducible factor 1α (HIF–1α), implicated in the progression of atherosclerosis. Ouyang et al. demonstrated that in a culture of human macrophages, the protein level of HIF–1α, induced by hypoxia, is reduced in a concentration-dependent way by treatment with curcumin (20–40 µM), which acts in a proteasome-dependent manner without affecting the mRNA level of the factor [113]. Moreover, it has been shown that curcumin causes the inhibition of the expression of HIF–1α through ERK signaling pathway. Following HIF–1α inhibition, curcumin also affects macrophage apoptosis induced by hypoxia; also, it significantly reduces the hypoxia-induced release of proinflammatory cytokines IL–6 and TNF–α from macrophages [113].

Curcumin has been evaluated under the aspect of AGEs inhibition, a process that is strictly related to arterial senescence and degeneration. A review has recently taken into account this issue [114]. Curcumin has been shown to reduce in vitro the generation of AGEs, as well as to counteract in vivo AGE formation. The possible mechanisms involved include methylglyoxal trapping, but also the capacity to prevent AGE formation, due to the antioxidant activity and clearance of free radicals. Curcumin can also act on the expression of AGE receptors by suppressing RAGE (a receptor that mediates the detrimental effects of AGEs) and enhancing AGE–R1 (a distinct receptor that mediates the detoxification of AGEs) [115]. It should be mentioned that curcumin may act as favorable modulator of PAD also influencing several pathways linked to DM, the disease whose complications often evolve toward vascular manifestations. A recent overview has considered by bioinformatic methodology the many targets that can be affected by curcumin due to its antioxidant and anti-inflammatory properties, which can modulate the disease’s progression [116].

#### 3.2.3. Activity: Clinical Studies

From a strict, clinical point of view, the data on the effectiveness of curcumin in PAD are limited. A meta-analysis of 5 randomized clinical trials conducted on a total of 192 healthy subjects with doses of curcumin ranging from 25 to 200 mg, showed that curcumin supplementation is able to improve vascular function, determined as the flow-mediated dilation of the brachial artery [117]. Alidadi et al. reported 7 randomized clinical trials conducted on humans, ranging from healthy subjects to obese people and type 2 diabetic patients, with a sample size ranging from 21 to 88 subjects [118]. Curcumin doses ranged from 150 mg/day to 2 g/day, over periods of 8–16 weeks. In healthy subjects, curcumin appeared to improve endothelial function, determined by a non-invasive ultrasound-based technique that measured flow-mediated dilation. On the contrary, in pathological conditions, curcumin treatment did not show an appreciable improvement in endothelial function, also due to the limited number of patients evaluated [118].

Despite its interesting biological properties, curcumin has the limitation of being poorly absorbed by the gastrointestinal tract due to its high hydrophobicity [119]. Several formulations were investigated to enhance curcumin bioavailability [120,121], in particular micro- and nano-formulations. An optimized curcumin formulation was developed and investigated [122]. It consisted of poly (propylene sulfide) (PPS) microparticles used for curcumin encapsulation and the delivery of the active principle on demand at the site of oxidative stress. An oil-in-water emulsion was obtained, which permitted, in the presence of ROS, the transition of hydrophobic PPS into more hydrophilic sulfoxide and sulfone, thus releasing curcumin. Following an in vitro evaluation of the system, the authors tested the formulation in vivo (curcumin-loaded PPS microspheres: 5 mg/kg of curcumin with 10.3 mg/kg of PPS, administered i.m. in the gastrocnemius and adductor muscles) in a streptozotocin-induced mouse model of DM in which hindlimb ischemia was induced [122]. The results show that curcumin microspheres were able to selectively reduce ROS levels in the tissue of the ischemic limbs, significantly increasing blood perfusion and the length of the vasculature, providing evidence for a favorable action of curcumin in the treatment of diabetic PAD. Another more recent attempt to produce “on demand” treatment for PAD has been reported [123]. It consists of PVAX (vanillyl alcohol-incorporated copolyoxalate) nanoparticles loaded with curcumin that release the drug at a pathologically elevated level of H_2_O_2_, as occurs at an ischemic site. After optimizing in vitro the formulation of curcumin–PVAX nanoparticles and demonstrating their antioxidant, anti-inflammatory and angiogenic activities, the authors evaluated the activity of the product in a mouse model of unilateral hindlimb ischemia. The in vivo results, obtained with the i.m. injection of curcumin–PVAX nanoparticles into the ischemic area on days 0, 3 and 7, showed significant blood-flow recovery and neovascularization, with almost complete blood-flow restoration 12 days after ischemia. The histological examination demonstrated the suppression of tissue damage and inflammatory responses, with an extensive expression of angiogenic factor CD31 [123]. Several other formulations of curcumin, including nano-formulations or encapsulation, have appeared on the market to ensure the optimal intake of the active compound [124,125].

#### 3.2.4. Safety Profile

Regarding curcumin toxicity, the compound is considered safe and has a good tolerability profile in humans [126,127,128]. The reported side effects mainly include gastrointestinal disturbances. However, in Italy, several cases of acute hepatitis have been reported to be associated with formulations that provide high bioavailability and high doses of curcumin [129]. In fact, the availability of many different formulations of curcumin with an enhanced bioavailability profile has focused on the question of safety. For instance, Fuloria et al. considered the curcumin toxicity profile that emerged during the most recent preclinical and clinical studies; excessive amounts of curcumin may lead to alterations in testosterone levels in men, influence blood clotting and contrast iron absorption [125].

In VigiBase curcumin has 71 potential side effects [95], such as gastrointestinal disorders (i.e., diarrhea and nausea), nervous system disorders (e.g., asthenia and others) and hepatobiliary disorders (e.g., acute liver failure, increase in alanine aminotransferase and aspartate aminotransferase, and others). More trials are needed to understand the safety of curcumin for pharmacological use, also reducing the number of additives, and considering customized microencapsulation [125].

### 3.3. Mangiferin

#### 3.3.1. Chemistry and Sources

Mangiferin is a natural C-glucoside xanthone (2-β-d-glucopyranosyl-1,3,6,7-tetrahydroxy-9H-xanthen-9-one) contained in many plant species, but especially in the fruit, kernels and the leaves of the mango tree (*Mangifera indica* L.), a native plant of India [130,131]. The mango tree also contains similar phenolic components, isomangiferin and homomangiferin, which contribute to the beneficial effects of the plant extracts [132,133]. Mangiferin has four hydroxyl groups in the xanthone nucleus and, thus, can be considered among the occurring phenolic compounds present in higher plants providing antioxidant activity.

#### 3.3.2. Activity: Preclinical Studies

Mangiferin has been reported to be an antidiabetic, anti-inflammatory, antimicrobial, immunomodulator, anticancer and hypocholesterolemic agent [133]. It has high antioxidant activities due to its hydroxyl groups and redox-active aromatic system of the catechol moiety [131,134]. In addition to the described scavenger activity on ROS, mangiferin can modulate the expression of several genes involved in inflammation and apoptosis, including the induction of the antioxidant Nrf2 pathway [135,136]. Furthermore, it can protect mitochondrial membranes against lipid peroxidation and prevent hydroxyl radical formation by inhibiting Fenton-type reactions [137]. Mangiferin exhibits a strong inhibition of oxidative stress associated with the endoplasmic reticulum by reducing ROS production and attenuating inositol-requiring enzyme 1 (IRE1) phosphorylation [138]. Mangiferin has also been shown to be an inhibitor of the NF–κB signaling pathway [131]. Furthermore, it has been evaluated as a possible pharmacophore structure for the development of new compounds with pharmacological activity in multiple pathological conditions [134], possibly related to inflammation and DM.

In HUVECs, mangiferin (20 µM) counteracts paraquat-induced endothelium damage, preserving the p120–catenin protein level [139]. Furthermore, it protects endothelial cells from oxidative injury induced by H_2_O_2_ or glycated protein–iron chelate, suggesting a protective role against pathologies linked to oxidative stress [140]. In an experimental model of high glucose/hypoxia-induced angiogenesis, mangiferin was effective in inhibiting angiogenesis by reducing hypoxia-inducible factor–1α, vascular endothelial growth factor and matrix metallopeptidase (MMP)–2 and MMP–9 [141]. Instead, Daud et al. observed that mango extract and mangiferin stimulated the migration of bovine endothelial aortic cells in a modified Boyden chamber assay, suggesting a role for the polyphenol in the promotion of the formation of new blood vessels [142].

The rationale of mangiferin use in DM is also related to its inhibition of AGE formation, hence counteracting the vascular damage typical of the disease. In a rat model of streptozotocin-induced diabetes, Hou et al. observed a sustained suppression of AGE production and a decrease in the protein expression of RAGE receptors; another relevant effect of mangiferin was the inhibition of NF–κB with a reduction in inflammatory cytokines [143]. Using an animal model of a mouse fed with a high-fat diet, Jiang et al. observed that mangiferin (5–20 mg/kg) could reduce plasma lipids and aorta wall thickening [144]. In oxidized-LDL-induced HUVEC injury, the compound was able to alleviate cellular dysfunction, reducing ROS levels, increasing the release of NO^•^ and activating the PTEN/Akt/eNOS signaling pathway [144]. Recently, a meta-analysis considering 19 studies on diabetic animals, mainly rats and mice, showed that mangiferin intake up to 422 mg/kg reduced blood glucose levels in a dose-dependent manner [145].

An extract of *Mangifera indica* administered to LDL-receptor-deficient mice for 2 weeks produced a reduction in plasma and liver cholesterol levels, ROS production in spleen mononuclear cells and increased plasma total antioxidant capacity [146]. Mangiferin activity in PAD was also indirectly investigated using ethanolic extracts of mango seed (EEMI) in an acute hindlimb ischemia-reperfusion model [147]. Streptozotocin-treated diabetic rats underwent a femoral artery ligation and, then, received EEMI (0.2–0.4 g/kg) for 14 days. Blood flow was observed to be significantly higher in treated animals than in the controls. The plasma levels of malondialdehyde, IL–6, TNF–α and IL–1β were reduced, while glutathione and IL–10 levels were increased in EEMI-treated animals, suggesting anti-inflammatory modulation [147]. In general, given the above reported experimental data, mangiferin appears to offer promise in the prevention and treatment of vascular disease also linked to diabetes and dyslipidemia.

#### 3.3.3. Activity: Clinical Studies

A 12-week, double-blind, placebo-controlled clinical study was conducted on 104 overweight patients with hyperlipidemia subdivided into 2 groups, administering a dose of 150 mg/day of mangiferin or placebo for 12 weeks. Treatment with mangiferin produced a significant decrease in the serum levels of triglycerides and free fatty acids, and the insulin-resistance index [148]. However, clinical trials conducted on the use of mangiferin in cardiovascular diseases and, moreover, in patients with PAD are lacking.

#### 3.3.4. Safety Profile

Mangiferin in high concentrations may cause damage to the mitochondrial respiratory chain, since free radicals are also needed for normal cellular activity [131]. The oral administration of 0.9 g of mangiferin has been reported to be harmless to adults [131,149]. Mangiferin per se did not have any adverse reports in VigiBase [95]; otherwise, *Mangifera indica* has 62 reports of unwanted effects, entirely from the Americas [95]. The side effects were mainly gastrointestinal (nausea and diarrhea), nervous system (dizziness) and skin (pruritus and erythema) disorders [95]. Overall, at moderate doses, the compound has been reported to be safe for humans.

### 3.4. Quercetin

#### 3.4.1. Chemistry and Sources

Quercetin is pentahydroxyflavone (3,3′,4′,5,7-pentahydroxyflavone) having five hydroxyl groups belonging to the group of flavonols found in many fruits (e.g., apples, grapes, berries and citrus fruits) and vegetables (e.g., onions, broccoli and Italian chicory) [150], and widely used as a food supplement in various commercial products (generally, 500 mg, twice daily) for circulation, immune system function and respiratory function [151]. Natural derivatives of quercetin, such as isoquercetin and rutin, exist in natural sources, for instance, in onions and citrus foods [152]. However, the low solubility in water and limited bioavailability of quercetin have limited the medical use of the compound. Several enzymatically modified derivatives of quercetin have been obtained and investigated, mainly for their favorable bioactivity and bioavailability. Quercetin derivatives have also been obtained by using engineered *Escherichia coli* and other microorganisms [152]. The structure–activity relationships of quercetin and its derivatives have been recently investigated in detail [153], including the antioxidant, anti-inflammatory and antidiabetic properties.

#### 3.4.2. Activity: Preclinical Studies

Quercetin shows prominent antioxidant potential and is considered an effective free-radical scavenger mainly based on several in vitro studies [154,155,156]. The antioxidant activity of quercetin is believed to be linked to the regulation of GSH levels [150]; moreover, the hydroxyl groups on the side phenyl ring of the molecule can bind to amino acid residues of key enzymes, such as acetylcholinesterase and butyrylcholinesterase, both linked to oxidative properties. Quercetin also increases the levels of endogenous antioxidant enzymes, e.g., catalase, GSH peroxidase and superoxide dismutase. More specifically, considering the vascular effects, quercetin induces the vasodilation of isolated rat arteries [157,158,159] and, in vivo, antihypertensive effects on rats fed with a high-fat, high-sucrose diet [158] and in spontaneously hypertensive animals, without effect in normotensive animals [160]. Generally, this flavonoid showed antiangiogenic activity in several experimental models [161,162,163].

The protective effect of quercetin against the activation of ER stress was attributed to the upregulation of markers, such as the 78 kDa glucose-regulated protein (GRP78), a molecular chaperone, and the C/EBP-homologous protein (CHOP) in unresolved diabetic and experimental ER-stress conditions [164]. Furthermore, quercetin pretreatment decreased the expression of tunicamycin-induced ER-stress markers in HUVECs [164]. In another study, mitochondrial-targeted quercetin activities were observed to be a mechanism of protection against neurodegenerative diseases [165]. Quercetin has been reported to protect against hydrogen peroxide-induced pheochromocytoma cell neurodegeneration [166].

In general, quercetin has shown anti-inflammatory action in several experimental models, mainly through mechanisms that inhibit NF–κB and cofactors in the chromatin of proinflammatory genes [167]. Quercetin also increases glucose uptake from the blood by inducing the glucose transporter GLUT4, and promotes glucose storage by the liver [156,168].

#### 3.4.3. Activity: Clinical Studies

Few trials have examined the effect of quercetin supplementation on blood pressure, with discordant results; however, generally, a reduction in systolic blood pressure both in healthy subjects and hypertensive patients has been reported [169,170,171,172]. Recently, a meta-analysis showed significant s in both systolic and diastolic blood pressure with quercetin supplementation; the treatments ranged from 4 to 10 weeks, with doses ≥500 mg/day [173]. Another meta-analysis showed a significant reduction in blood pressure and, furthermore, for participants receiving quercetin for at least 8 weeks, a decrease in triglycerides in trials with a parallel design [174]. A double-blinded, placebo-controlled cross-over study conducted on 93 overweight or obese subjects aged 25 to 65 years with metabolic syndrome traits showed that quercetin administered at 150 mg/day (6-week-treatment period) reduced systolic blood pressure and plasma oxidized-LDL levels in overweight subjects with a high-CVD-risk phenotype [171].

In the scientific literature, no clinical studies have been reported on the use of quercetin in the treatment of PAD.

#### 3.4.4. Safety Profile

Egert et al. reported that daily supplementation with 150 mg of quercetin/day, administered orally to volunteer subjects with a high-CVD-risk phenotype for 6 weeks, was safe [171]. In general, the clinical studies reported that the orally administered quercetin use was safe and well-tolerated [173,174].

Quercetin per se has few adverse effects reported in VigiBase [95]; a total of 45 unwanted effects are described, mainly in the Americas and Europe. Nervous system disorders (e.g., dizziness and headache), respiratory disorders (e.g., dyspnea), pruritus and drug interaction are the most recurrent effects [95]. Nevertheless, in general, moderate doses of quercetin are described as safe.

### 3.5. Resveratrol

#### 3.5.1. Chemistry and Sources

Resveratrol is a stilbene derivative (3,5,4′–trihydroxystilbene) occurring both as *trans*- and *cis*-isomers that are present in a variable percentage in several natural sources, the *trans*- form being the most abundant and mainly responsible for the cardiovascular effects [175]. Resveratrol is a phytoalexin (a class of antimicrobials synthesized by plants under pathogen infection) found in many plant foods. Its presence is well-known in red grapes (skin) and red wine, but even in tea and berries and also in various medicinal plants (e.g., *Polygonum* spp. roots) used in popular medicine as treatments for allergic and inflammatory diseases [176].

#### 3.5.2. Activity: Preclinical Studies

Resveratrol has antioxidant and free-radical scavenger activities that may be responsible for its several biological activities, such as anti-inflammatory, antiatherosclerosis and anticarcinogenic effects [177,178]. The structural determinants of the antioxidant activity of resveratrol have been linked to the presence of hydroxylic functions, in particular, but not only, the hydroxyl group at 4′ position, as demonstrated by an investigation on the derivatives [179].The dose-dependent biphasic hormetic effects of resveratrol have been reported: at low concentrations, it acts as an antioxidant that protects tissues from oxidative stress, while at high concentrations, it may be a pro-oxidant that increases oxidative stress [180]. Similarly, low and high concentrations can provide chemoprevention or cytotoxicity, respectively, against cancer cells [181]. However, resveratrol is especially known for its beneficial effects in cardiovascular diseases. Several authors have shown that it causes vasodilation in different types of isolated arteries obtained from various animal species (e.g., guinea pigs, pigs, rats and sheep) [182,183,184,185]. Studies showed that the anti-inflammatory activity of resveratrol is mainly mediated by antiadrenergic and antiprostaglandin activation [176]. Resveratrol reduced the sensitivity of myofilaments to free calcium in vascular smooth muscles and enhanced acetylcholine-stimulated calcium increase in the endothelium, promoting NO^•^ production and thus vasorelaxation [182]. At nanomolar concentrations, it induces the endothelial production of NO^•^ by activating the estrogen receptor-α (ERα)–Cav–1–c–SRC interaction, resulting in NO^•^ production through a Gα–protein-coupled mechanism [186]. It down-regulates VEGF/fetal liver kinase-1 (Flk–1) (VEGF receptor-2) expression and, therefore, modulates hyperpermeability and junction disruption in glomerular endothelial cells. In addition, resveratrol ameliorates high-glucose-induced hyperpermeability mediated by overexpressed caveolin-1 in aortic endothelial cells [187]. A recent study using a palmitate-induced insulin-resistance model revealed that resveratrol suppresses IKKβ/NF–κB phosphorylation, TNF–α and IL–6 production, and restores the IRS–1/Akt/eNOS signaling pathway in endothelial cells [188]. Resveratrol has been reported to block the TNF–α-induced activation of NF–κB in coronary arterial endothelial cells and inhibit inflammatory mediators [189], exerting the effect through the action on the IKK cascade, attributing to this mechanism its antioxidant properties. The report by Kim et al. demonstrated that resveratrol, as well as hesperidin and naringenin, reduces high-glucose-induced ICAM–1 expression via the p38 MAPK signaling pathway, contributing to the inhibition of monocyte adhesion to endothelial cells [190]. Recently, resveratrol showed an inhibitory effect against NF–κB p65 and proinflammatory mediators, including TNF–α, ICAM–1 and MCP–1 in endothelial cell lines [191]. Resveratrol confers a protective effect against high-glucose-induced oxidative stress in endothelial cells and vascular protection in high-fat-diet mice, through the Nrf2 pathway [192].

In several experimental models, in vitro and in vivo, resveratrol improved glucose homeostasis and insulin sensitivity [193,194]. An in vivo study conducted on diabetic rats demonstrated that the compound elicits antidiabetic potential by stimulating intracellular glucose uptake and the modulation of sirtuin-1 activity [195]. Resveratrol also relieves the status of diabetic nephropathy, kidney and oxidative stress in diabetic rats [196]. Resveratrol inhibits ATP-dependent K^+^ channels and voltage-dependent delayed-rectifier K^+^ channels in β-cells, suggesting its beneficial role in delaying the onset of insulin resistance and improving insulin secretion [197].

#### 3.5.3. Activity: Clinical Studies

Resveratrol is mainly known since it offers a possible explanation for the so-called “French paradox”, which is the low frequency of heart disease in the French population despite the relatively high-fat dietary use, believed to be linked to red wine consumption, as a source of resveratrol [198]. Several clinical trials considered resveratrol both in healthy volunteers and patients with various cardiovascular diseases, but they all considered a limited number of participants. Patients with coronary artery disease received 10 mg/day of resveratrol for 3 months, showing an increase in flow-mediated vasodilation and, in general, an improvement in cardiovascular parameters [44]. A clinical study using resveratrol revealed that oral, 100 mg consumption for 12 weeks may support the prevention of cardiovascular disease and atherosclerosis by stimulating endothelial function [199]. Furthermore, resveratrol also modulates NO^•^ metabolism and contributes to improved vascular function in hypertensive and dyslipidemic patients [200]. In addition, a cross-over, double-blind, placebo-controlled study in which 22 healthy adults received 250 and 500 mg of resveratrol revealed dose-dependent increases in cerebral blood flow [201]. Another study also showed that the acute administration of 75 mg of resveratrol increased neurovascular coupling and cognitive performance in 36 subjects affected by T2DM, improving cerebral perfusion [202]. A meta-analysis concerning 3 clinical trials for a total of 50 DM subjects treated with resveratrol at doses of 10, 150 and 1000 mg daily, for a period of 4 to 5 weeks, did not show any favorable effects on the glycosylated hemoglobin A1c level or on insulin resistance [203]. Recently, a double-blind and randomized clinical trial, entitled “Resveratrol to Improve Outcomes in Older People With PAD” (RESTORE), was conducted on 66 patients with PAD treated with 125 mg/day or 500 mg/day of resveratrol, or placebo for 6 months [204]. However, the trial did not show reliable confirmation that resveratrol improves walking performance detected by the 6min walk test among patients with PAD [204]. Other clinical studies enrolling a higher number of patients affected by PAD are required to evaluate the efficacy of resveratrol on this disease.

#### 3.5.4. Safety Profile

Resveratrol exhibited the systemic inhibition of P450 cytochromes when taken in high doses [205]. Furthermore, the ingestion of 25 mg/kg of resveratrol for 60 days in rats altered their thyroid function, causing a goitrogenic effect [206]. Other studies using oral doses of 200 mg/kg/day in rats and 600 mg/kg/day in dogs did not report adverse effects [207]. Few clinical trials have evaluated the safety and tolerability of resveratrol; Brown et al. reported gastrointestinal discomfort at doses of 2.5 and 5 g/day [208]. The administration of resveratrol with single doses of 0.5, 1, 2.5 or 5 g orally in 40 healthy volunteers caused minor adverse events that resolved spontaneously in a few days [209]. In general, 150 mg/day for adults is accepted to be safe [210].

Resveratrol per se has very few adverse reports in VigiBase [95]; a total of 20 unwanted effects have been described in the Americas, Europe and Oceania. Among these, the most common are general disorders (e.g., malaise and fatigue) and disorders of the gastrointestinal and nervous systems [95]. Therefore, low doses of resveratrol could be considered safe and potentially useful for vascular disorders.

## 4. Conclusions

Polyphenols have been extensively investigated for their beneficial effects on many clinical conditions, as well as cardiovascular-related diseases. Despite the profusion of experimental data suggesting potential mechanisms favorable for clinical use, trials on the side of peripheral vascular diseases are still hindered by the absence of large studies with sufficient statistical power in order to demonstrate the true efficacy of the compounds. The requirement of large resources to program a clinical study that does not necessarily allow for a unique patent outcome often discourages an industrial gamble, reducing the real possibility of developing new drugs in therapy. Resources for research on natural products from non-profit organizations are necessary.

Emerging polyphenolic compounds were selected based on their appearance in the literature and have been discussed here, with a focus on their role in improving endothelial and, in general, vascular functions; molecular mechanisms and preclinical and clinical evidence were highlighted. Overall, the data collected in this review suggest the potential role of selected polyphenols in the treatment of PAD, also concomitant with DM; however, clinical data on efficacy are still limited and need to be improved. Despite the extensive preclinical experimental results that confirm the potential role of baicalein, curcumin, quercetin, mangiferin and resveratrol against vascular diseases, their clinical effectiveness is still only preliminarily demonstrated, although there is evidence of safety for the selected doses. Natural-drug compounds that provide significant biological activities in the specific vascular district have been suggested, and the task of a chemical approach could be directed to optimize their bioavailability. However, caution should be devoted to the fact, as already observed that, for example, with curcumin, an absolute increase in bioavailability does not necessarily mean an improved benefit/risk ratio when the product is introduced for clinical use. Moreover, derivatives or analogs obtained on the structural basis of natural products do not necessarily produce to compounds that are safe for human use. The occurrence in the molecule of phenolic hydroxyl groups also raises the question of biological stability as drugs. Therefore, a new concept of the wise use of available natural resources should be applied, also in the prospective development of drugs, keeping in mind the limits and advantages of medicinal plants and relative natural compounds.

As a general consideration, a diet rich in polyphenols reduces the risk of cardiovascular adverse events, including PAD. Polyphenols, together with adequate moderate aerobic exercise, can help prevent and reverse age-associated arterial stiffness. In fact, exercise therapy is considered a class IA (highest level) recommendation for the treatment of patients with PAD. Unfortunately, the long-term participation in perspective clinical and population-based programs is scarce, and therefore this approach is still just outlined. The use of polyphenols, both as dietary intake and dietary supplements, could represent a favorable approach to maintaining the integrity of peripheral blood vessels and limiting the harmful effect of oxidants. The present evidence suggests the validity of further clinical trials to define the role of this class of compounds in the prevention and treatment of vascular artery disease.

**Table 1 molecules-27-07110-t001:** Plant-derived polyphenols with potential activity in preventing or improving cardiovascular diseases and, thus, also PAD: in vitro and in vivo studies using various cellular or animal models.

Chemistry	Natural Compounds	Food Sources/Medicinal Plants	Experimental Models	Findings	Cellular Mechanisms	References
Flavones	Baicalein	- *Scutellaria baicalensis* - *Scutellaria lateriflora* - *Oroxylum indicum*	- EA.hy926 cells - HUVECs - Mouse	- Improves vascular permeability and function - Inflammation inhibition	- Decreases AGEs and TNF–α - Decreases NF–κB activation - Increases AMPK activation - Decreases CAMs expression - Decreases MCP–1 and IL–8 - Decreases H_2_O_2_	[68,70,71,75,77,78,79,80,81,82,83,84,85,86,87,88,89,90,91,92]
Diarylheptanoids	Curcumin	- *Curcuma longa*	- Mesenteric arteries - Rat- Mouse(limb ischemia)	- Vasorelaxation - Blood-flow recovery - Vascular inflammation inhibition - Decreases hyperglycemia	- Decreases O_2_^●−^ - PCK inhibition - Decreases ICAM–1 and NOX2 - Decreases ROS - Counteracts AGE formation - HIF–1α inhibition	[100,101,106,107,108,109,110,111,112,113,114,115,122,123]
Xanthones	Mangiferin	- *Mangifera indica* - *Anemarrhena asphodeloides* - Ferns	- HUVECs - RRCECs - Mice - Diabetic mice - Diabetic rats	- Improves endothelial function - Promotes new blood vessel formation - Improves blood flow - Antihyperglycemic effect - Inflammation inhibition	- Reduces ROS - Inhibits NF–κB activation - Inhibits HIF–1α - Counteracts AGEs - Cytokine inhibition - Inhibits endoplasmic reticulum stress	[135,136,137,138,139,140,141,142,143,144,145,146,147]
Flavonols	Quercetin	- Apples - Berries - Grapes - Onions - Tea - Red wine	- Collagen glycation - Glucose autoxidation - HAECs - Aortic rings (various animal species)	- Vasorelaxation - Glycation inhibition - Inhibition of endothelium dysfunction - Inflammation inhibition	- Increases eNOS - Decreases AGEs - Activation of AMPK - Inhibition of endothelin–1 - Inhibition of phosphodiesterase - Decreases COX2 expression and lipoxygenase	[150,154,155,156,157,158,159,160,161,162,163,164,165,166,167,168]
Stilbenes	Resveratrol	- Berries - Blueberries - Cocoa - Nuts - Peanuts - Pomegranates - Red grapes - Red wine - Tea	- HUVECs - VSMC - Rat aortic rings - Mesenteric arteries - BSA assay - α–amylase - α–glucosidase - Diabetic rat retinas	- Vasorelaxation - Improves endothelial function - Reduces platelet aggregation - Inhibition of AGE formation - Inhibition of cell proliferation - Glycation inhibition - Antihyperglycemic effect	- cGMP increase (dSCA) - cAMP phosphodiesterase inhibition - Increases NO^•^ production - AMPK activation - SIRT1 activation - Scavenges ROS - Increases ERα–Cav–1–c–SRC interaction - Inhibits AGE-induced TGF–β1 mRNA increase - Trapping MGO - Inhibition of α–amylase and α–glucosidase activities - SIRT1 activator - Increases tight-junction protein occludin	[176,178,179,180,181,182,183,184,185,186,187,188,189,190,191,192,193,194,195,196,197]

AGEs, advanced glycation end-products; AMPK, adenosine monophosphate-activated protein kinase; CAMs, cell adhesion molecules; GO, glyoxal; HIF–1α, hypoxia-inducible factor 1α; HAECs, human aortic endothelial cells; HRPE cells, human retinal pigment epithelial cells; HUVECs, human umbilical vein endothelial cells; IL–6 or IL–8, interleukin–6 or –8; ICAM–1, intercellular adhesion molecule 1, MCP–1, monocyte chemoattractant protein–1; MGO, methylglyoxal; NOX2, phagocytic NADPH oxidase, NO, nitric oxide; eNOS, endothelial nitric oxide synthase; O_2_^●−^, superoxide anion; VSMCs, vascular smooth-muscle cells; dSCA, denuded-sheep coronary arteries; PKC, protein kinase C; RRCECs, rat retinal capillary endothelial cells; SIRT1, silent mating- type information regulation 2 homolog 1.

**Table 2 molecules-27-07110-t002:** Plant-derived polyphenols with potential activity in preventing cardiovascular diseases and, thus, also PAD: clinical trials and adverse reactions.

Polyphenols	Clinical Trials	Systematic Review and Meta–Analysis	Findings	References
Baicalein	- Safety and PK evaluation in randomized, double-blind, placebo-controlled clinical trials (36–110 subjects)	– –	- No clinical data available on activity - Well-tolerated, no serious adverse effects	[95,96,97]
Curcumin	- Randomized clinical trials on healthy, obese or DM-affected subjects	- A meta-analysis on 192 healthy subjects	- Vascular-function improvement in healthy subjects - Modest effect on obese people - Good tolerability profile - Sporadic cases of hepatitis	[11,95,118,125,126,127,128,129]
Mangiferin	- Randomized controlled trial on hyperlipidemic subjects	– –	- Reduces triglycerides and free fatty acids, and insulin resistance index - Reported as safe in humans (few side effects with *Mangifera indica* extracts)	[95,148,149]
Quercetin	- A double-blinded, placebo-controlled cross-over study, on 93 overweight or obese subjects - Randomized controlled cross-over trial on 30 healthy subjects	- Meta-analysis on 841 normal and hypertensive subjects	- Reduces systolic blood pressure in healthy and hypertensive subjects - Reduces systolic blood pressure and plasma ox–LDL in hypertensive overweight subjects - No effect on heart disease risk factors - Reported as safe in moderate doses	[95,169,170,171,172,173,174]
Resveratrol	- Several clinical studies on healthy subjects and hypertensive and dyslipidemia or DM-affected patients - One pilot randomized clinical trial of 66 participants with PAD	- Meta-analysis on 50 DM subjects	- Improvement in cardiovascular parameters - Flow-mediated vasodilation - No clinical improvement in 6-minute walk test - No improvement of glycosylated - hemoglobin A1c levels and insulin resistance - Few side effects (gastrointestinal effects)	[44,95,199,200,201,202,203,204,205,206,207,208,209,210]

## Figures and Tables

**Figure 1 molecules-27-07110-f001:**
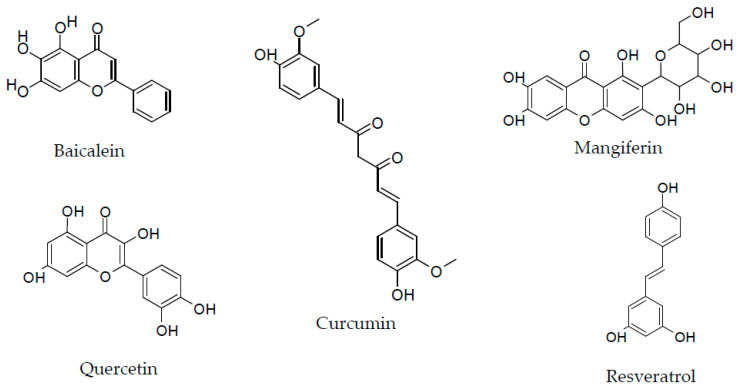
Chemical structure of the considered polyphenols.

## Data Availability

Not applicable.

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
