# Peer review of "Selected Plant-Derived Polyphenols as Potential Therapeutic Agents for Peripheral Artery Disease: Molecular Mechanisms, Efficacy and Safety"

_molecules, 2022, doi:10.3390/molecules27207110_

Round 1

Reviewer 1 Report

This manuscript reviewed recent advances on the effect of natural polyphenols in the pathogenesis of vascular diseases and periphery artery disease (PAD) with particular emphasis on five polyphenols including baicalein, curcumin, mangiferin, quercetin, and resveratrol. This manuscript is very interesting and well-written. The summarization of the molecular mechanisms and preclinical and clinical studies of each polyphenol would contribute to a better understanding of natural polyphenols on PAD. In addition, the safety profile and the unwanted effects were also outlined in this manuscript. I suggest accept in the present form. 

Author Response

The authors thank the reviewer for the appreciation for their article and for the positive judgment.

Reviewer 2 Report

The topic of article is interesting. Characterization of five polyphenols such as baicalein, curcumin, mangiferin, quercetin resveratrol provided valuable information but not sufficient. The potential impact of polyphenols in peripheral artery  disease seems very important.  Therefore, the undertaken research topic by Authors is very interesting. However, the molecular mechanisms are not described well and Authors didn’t connect antioxidant activity od selected polyphenols with their chemical structure.

The reviewer suggests major revisions. The list of suggestions and remarks are below:

1.       In part 1 Authors should describe excluding and including criteria of searching. (page 4)

2.       In part 2, section 2.1. Authors should add more studies, so proper literature should be provided.

3.       In all part 3 - title: “Chemistry” should be changed on “Chemistry and sources”

4.       In section 3.1.2 the first sentence …baicalein and baicalin produce many pharmacological effects… should be refreshed.

5.       I section 3.1.2 the mechanism of action should be better described. Authors should connect chemical structure of mentioned polyphenols with antioxidant activity. Moreover, Authors in line 275 should describe mechanism.

6.       In section 3.1.3 Authors should add conclusion of clinical study (line 319)

7.       In section 3.2.2 Authors should introduce some words to describe studies carried out on animal model (line 355)

8.       In section 3.3 sentence in line 478-479 should be refreshed.

9.       In section 3.3.2 Authors should connect chemical structure with antioxidant activity.

10.   In section 3.4.1. Authors should indicate rich sources of quercetin.

11.   In section 3.4.2 there is no link between the chemical structure and the antioxidant activity.

 In Addition, mechanism of quercetin action should be better describe.

12.   The above-mentioned comments also are applied to section 3.5.3

13.   Conclusion should be refreshed. Authors should underline impact of selected polyphenols in treatment of PAD.

I would like to underline that Authors summed up information in tables , what is suitable for review work and allow to present valuable information in legible way.

Author Response

The authors thank the reviewer for comments and suggestions to improve the manuscript. We have implemented information regarding molecular mechanisms of action of the five compounds in order to better explain the structure-activity relationship, as described in the literature.

  1. In part 1, the including and excluding search criteria have been reported. Furthermore, the authors chose not to write a systematic review, but a general review to be free to include various items also from other searches.
  2. More studies have been provided in Part 2, Section 2.1.
  3. In all Part 3, as suggested, the subtitle “Chemistry” was changed to “Chemistry and sources”
  4. In Section 3.1.2 – The first sentence has been written more clearly, and an additional bibliographical entry has been added.
  5. As suggested, the mechanism of action has been better described, also with respect to links to the chemical structure of polyphenols with antioxidant activity. At line 312 the mechanism of baicalein through the TLR4 pathway has been reported.
  6. In Section 3.1.3 the conclusion related to the clinical study has been added.
  7. In Section 3.2.2 the ischemia and reperfusion injury animal model has been introduced.
  8. In Section 3.3 the sentence has been rewritten.
  9. In Section 3.3.2 more details regarding the chemical structure and antioxidant have been provided.
  10. In Section 3.4.1 rich sources of quercetin, as well as enzymatically derived analogues, have been reported.
  11. In Section 3.4.2 the chemical structure and antioxidant mechanisms of quercetin have been presented.
  12. The above-mentioned comments have been also considered for the following section.
  13. The conclusions have been improved.

The authors thank the reviewer for the suggestions helping to improve the manuscript.

Reviewer 3 Report

Dear Authors

Thank you for submitting of your revision paper.

The purpose of this review was to take into account the mechanisms that lead to endothelium dysfunction, such as the glycoxidation process and the production of ad18 vanced glycation end products (AGEs) that result in protein misfolding, and to suggest plant-de19 rived polyphenols that could be useful in PAD
After careful evaluation of the manuscript title "
 SELECTED PLANT-DERIVED POLYPHENOLS AS POTENTIAL THERAPEUTIC AGENTS FOR PERIPHERAL ARTERY DISEASE: MOLECULAR MECHANISMS, EFFICACY AND SAFETY", you can find below my comments. 
- The work is understandable, correct, and appropriate for the journal. 
- The purpose of and problems to solve in the work are clearly stated. 
- The purpose or goal of the work is within the journal’s scope. 
- The introductory section adequately explains the framework and problems of the research. 
- Importance and impact: the presented results are of significant importance and impact to advancement in the relevant field of research. 
- The manuscript is well written, has important message, and should be of great interest to the readers. 
-

The formulas in Figure 1 should be improved using a specific program

Overall, it is an important study, and should be considered for publication in Moleculese .
I recommand your research article for publication in the Molecules
  with minor revisions.
Best regards

Author Response

The authors thank the reviewer for the positive judgement on the present manuscript.

  • Figure 1 was improved by changing the format to make this more definite; the ChemSketch program was used. Please note that the pdf file, automatically generated by the system for the reviewing process, has a low image definition; so the image quality will appear better in the final publication form.

Reviewer 4 Report

Thank you very much for submitting your review article to MOLECULES. This ms is valuable as a review article for clinical application, and its usefulness is judged to be great. If there are clinical research trends and/or perspective based on ADMET or AI/big data, it is meaningful to the readers.

-Extensive editing of English language and style required.

-Please list  polyphenols applied in actual clinical practice. 

Author Response

- The manuscript was carefully examined and the English improved.

- Thanks for the interesting question. Unfortunately, the search in various databases (PubMed, Cochrane Library and Google Scholar) provides some studies on phenolic compounds as extracts, and not as isolated compounds. In general, from reviewing the literature and clinical guidelines, we have not found any phenolic preparations or phenolic compounds indicated in the clinical treatment of PAD based on meta-analyses. On the contrary, many products are sold as food supplements or nutraceuticals, without adequate evidence. For this, a list of polyphenols for clinical practice cannot be redacted. In this review, we have suggested five polyphenols that can be used, but other clinical studies are requested for an appropriate administration in patients.

Round 2

Reviewer 2 Report

I accept resubmitted version of manuscript.

Author Response

The authors greatly thank the reviewer for appreciation regarding the revised version of the manuscript.

Reviewer 4 Report

Thank you very much for your revision. 

The reviewer have carefully read your ms and raised some additional comments as below:

If there are no other clinical papers/reports on the designated polyphenolics, it would be helpful to the reader if the authors could describe the future perspective on the following: 1) Is there no sufficient medicinal efficasy? 2) Whether it is industrially worthless, 3) Do you need their derivatives or other chemically synthetic approaches? 4) Is a new concept for natural drug development strategy necessary?

Author Response

The authors thank the reviewer for the further suggestions to complete the revision process of the manuscript. According to the advices, the paragraph “Conclusion” was modified to deeper discuss our results, according to the reviewer’s proposals.